# Structure-Preserving 3D Garment Modeling with Neural Sewing Machines

Xipeng Chen[1], Guangrun Wang[2], Dizhong Zhu[2], Xiaodan Liang[1], Philip H. S. Torr[2], Liang Lin[1]*

[1] Sun Yat-sen University
[2] University of Oxford

chenxp37@mail2.sysu.edu.cn, wanggrun@gmail.com,
Amos.Zhu@hotmail.com, liangxd9@mail.sysu.edu.cn,
philip.torr@eng.ox.ac.uk, linliang@ieee.org

## Abstract

3D Garment modeling is a critical and challenging topic in the area of computer vision and graphics, with increasing attention focused on garment representation learning, garment reconstruction, and controllable garment manipulation, whereas existing methods were constrained to model garments under specific categories or with relatively simple topologies. In this paper, we propose a novel Neural Sewing Machine (NSM), a learning-based framework for structure-preserving 3D garment modeling, which is capable of learning representations for garments with diverse shapes and topologies and is successfully applied to 3D garment reconstruction and controllable manipulation. To model generic garments, we first obtain sewing pattern embedding via a unified sewing pattern encoding module, as the sewing pattern can accurately describe the intrinsic structure and the topology of the 3D garment. Then we use a 3D garment decoder to decode the sewing pattern embedding into a 3D garment using the UV-position maps with masks. To preserve the intrinsic structure of the predicted 3D garment, we introduce an inner-panel structure-preserving loss, an inter-panel structure-preserving loss, and a surface-normal loss in the learning process of our framework. We evaluate NSM on the public 3D garment dataset with sewing patterns with diverse garment shapes and categories. Extensive experiments demonstrate that the proposed NSM is capable of representing 3D garments under diverse garment shapes and topologies, realistically reconstructing 3D garments from 2D images with the preserved structure, and accurately manipulating the 3D garment categories, shapes, and topologies, outperforming the state-of-the-art methods by a clear margin.

## 1   Introduction

A long-standing vision in computer vision and graphics is generic 3D garment modeling, which can discover patterns in the clothing data, provide a better understanding of the clothing data, and learn reconstructable and manipulable representations for the clothing data. In the past decades, the dominant paradigm in 3D garment modeling resolves to physics-based simulators, which were computationally expensive and time-consuming [3, 35]. Recently, deep learning has been introduced and has made great strides [24, 49], allowing a wide range of applications, e.g., 3D virtual try-on [48], garment design [44], and virtual digital humans [1].

Despite achieving remarkable progress, prior works can only model specific categories or clothes with relatively simple topologies. Specifically, *template-based methods* [17, 49, 13, 12] use pre-defined templates with fixed topology to model garments for each category, which have difficulty

---

*Corresponding author is Liang Lin.

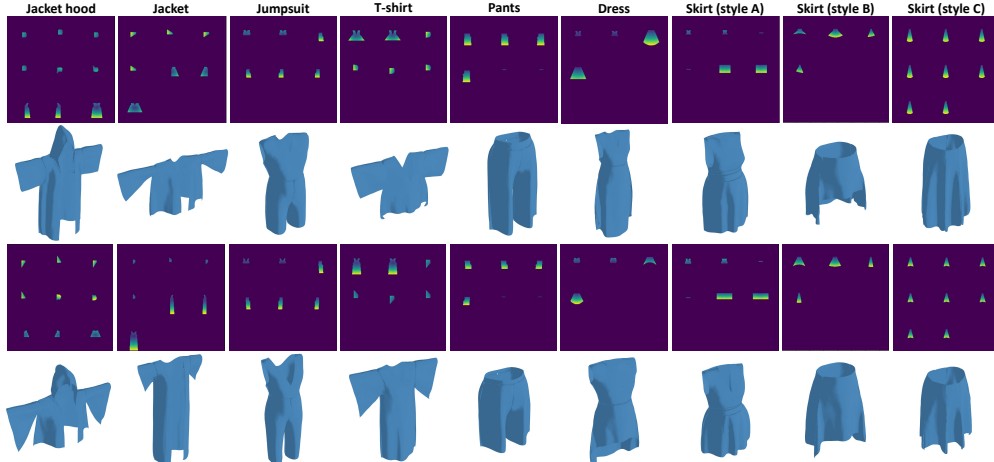

Figure 1: Illustration of our NSM for modeling garments with diverse shapes and topologies. The first and third rows denote the predicted UV-position maps with masks. The second and fourth rows denote the predicted 3D garment shape.

in representing real garments with diverse styles and categories. *SMPL-based methods* [21, 32, 33] use displacement on an SMPL model to represent different kinds of garments, assuming that the garments share the same topology with human bodies, but this assumption doesn't hold for generic garments, e.g., a skirt doesn't have legs (see Fig. 3). Recently, *UV-parameterization-based* methods [11, 28, 30, 45, 46, 17] convert a garment to a UV position map representation by storing the mesh shape and topology on a learned 2D map representation, but they use either pre-defined UV maps or rely on human body mesh parameterization. Without generic garment modeling, existing methods have shown inferior capability in 3D garment reconstruction and controllable garment manipulation.

In this paper, we propose a neural sewing machine (NSM), a learning-based approach for modeling generic garments with diverse shapes and topologies. Mimicking a physical sewing machine that is utilized for modeling realistic clothes, we define an NSM to model 3D garments using sewing patterns that are widely-used terms in physical garment sewing and production. A sewing pattern describes what atomic panels that a garment is composed of and how these atomic panels are sewn together (see Fig. 3). Such sewing patterns can not only truly reflect the garment model in the physical world but also uniquely describe the generic garments with diverse categories and shapes. With

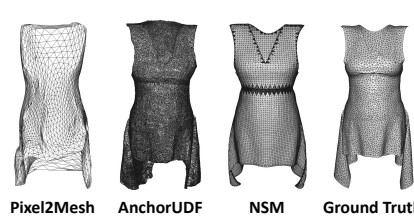

Figure 2: Topologies of 3D garments obtained from different methods (**Best view via zoom-in**).

this strong prior, our NSM approximates the garment surface by mapping the 2D atomic panels to the surface of the 3D shape, enforcing each warped panel to be as isometric as possible to the corresponding 3D garment (see Fig. 1), as opposed to previous methods that implicitly predict a 3D garment from a latent code without modeling explicit local structure [38, 47, 8, 30, 39]. Our NSM guarantees that the garment surface locally resembles the Euclidean plane and the mapping between the 3D garment and the 2D panels is isometric when only limited stretches exist on the garment surface (see comparisons in Fig. 2).

While the idea of an NSM is elegant, designing it faces multiple challenges. **First**, unifying the embeddings for generic garments with diverse categories, shapes, and topologies is challenging. To achieve this goal, we statistically analyze the panels of different garment categories in the dataset and construct basic panel groups, based on which we use PCA to build a parametric unified garment encoder. **Second**, mapping sewing pattern embeddings to 3D shapes is hard, which requires a warped panel to be isometric to the 3D garment. To solve this problem, we decouple 3D garments into sewing-pattern-based UV-position maps with masks and use an inversed PCA and a CNN to build a generic garment decoder. **Third**, existing deep models tend to filter out high-frequency signals and ignore modeling the seams between panels, preventing them from perceiving the 3D structure. To avoid this, we propose an inner-panel loss, an inter-panel loss, and a surface-normal loss to learn the structure-preserved representations for the 3D garments.

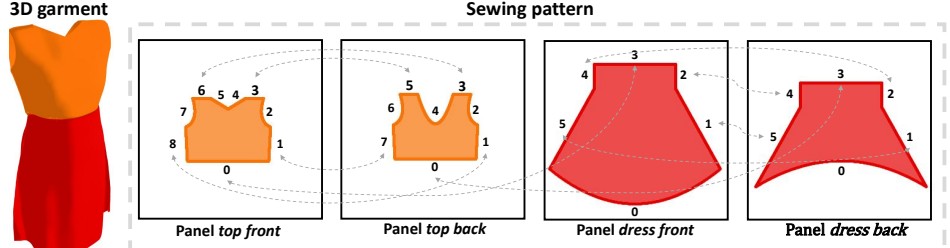

Figure 3: Definition of sewing pattern. A 3D skirt is composed of four panels, and each arrowed dashed line indicates the two edges from different panels are stitched together.

Thanks to our NSM, we can model generic garments with reconstructable and manipulable representations, allowing rich potential applications. For example, our NSM can arbitrarily edit the category and shape of garments as if it were a physical sewing machine. This interpretable manipulation is quite beneficial for garment design. As another example, our NSM can reconstruct 3D garments from 2D images with high reconstruction accuracy and high-quality 3D structure preservation.

In summary, this paper makes five non-trivial contributions.

- We propose an NSM, a novel 3D garment modeling method capable of modeling generic garments. Our NSM takes the sewing pattern as a solid prior, which can truly reflect the fidelity and diversity of clothes in the real world, so it can model the real garment structures and learn general garment representations.

- A unified sewing pattern encoder for generic garments with diverse categories, shapes, and topologies is proposed in our NSM, where we statistically construct basic panel groups for sewing patterns based on which we use PCA for encoding.

- A generic garment decoder for mapping sewing pattern embeddings to 3D shapes is proposed in our NSM, where we decouple the 3D garment shapes into sewing-pattern-based UV-position maps with masks and use an inversed-PCA and a CNN for decoding.

- Our NSM addresses the inability of deep models to achieve fidelity of 3D garment structure by proposing novel structure-preserving losses, including an inner-panel loss and an inter-panel loss.

- Extensive experiments show that our NSM achieves state-of-the-art results in multiple 3D garment modeling tasks on the large 3D garment dataset with sewing patterns, including representing 3D garments under diverse garment shapes and topologies, realistically reconstructing 3D garments from 2D images with the preserved structure, and accurately manipulating the 3D garment categories, shapes, and topologies.

## 2   Related Work

**Physics-based simulators.** Physics-based methods dating back to the 1980s required enormous computational effort due to the large number of triangular faces that need to be processed [31]. Later, to reduce the computational complexity, some works proposed efficient methods, such as using low-resolution simulators and small models [14, 36, 25], but searching for optimal hyperparameters for these simulators still required a lot of time. Later, to reduce the hyperparameter search time, some works proposed to learn, estimate, or speculate hyperparameters [22, 37, 29], but the controlled environment and simulator were indispensable conditions. Nevertheless, the complexity of physics-based simulators is high, with a lot of working time required.

**Template-based models.** An intuitive way to model 3D clothes is to use templates. Pioneeringly, Chen et al. [7] defined various garment part templates as a database and searched the required parts from the database to sew them into clothes. However, this searching-and-matching is impractical since the real world is dependent on infinite garment part templates. To solve this problem, parametric templates were proposed [17, 49, 13, 34, 5], in which templates of the same type but different sizes can be described by values. Although this significantly reduces the template number, designing parametric templates is still inefficient. Hence, some works proposed using human bodies as templates and added displacement to SMPL [20] (a typical human body model) [21, 6, 32, 33, 2]. However, these methods assume that garments and humans share a topology, which doesn't hold for generic garments, e.g., a skirt doesn't have legs. To make the templates more descriptive of real-world clothes, recently, sewing patterns have been considered promising templates [10, 43, 28, 16, 42, 40]. However, the use of sewing patterns is still similar to ordinary parametric templates, so it still faces

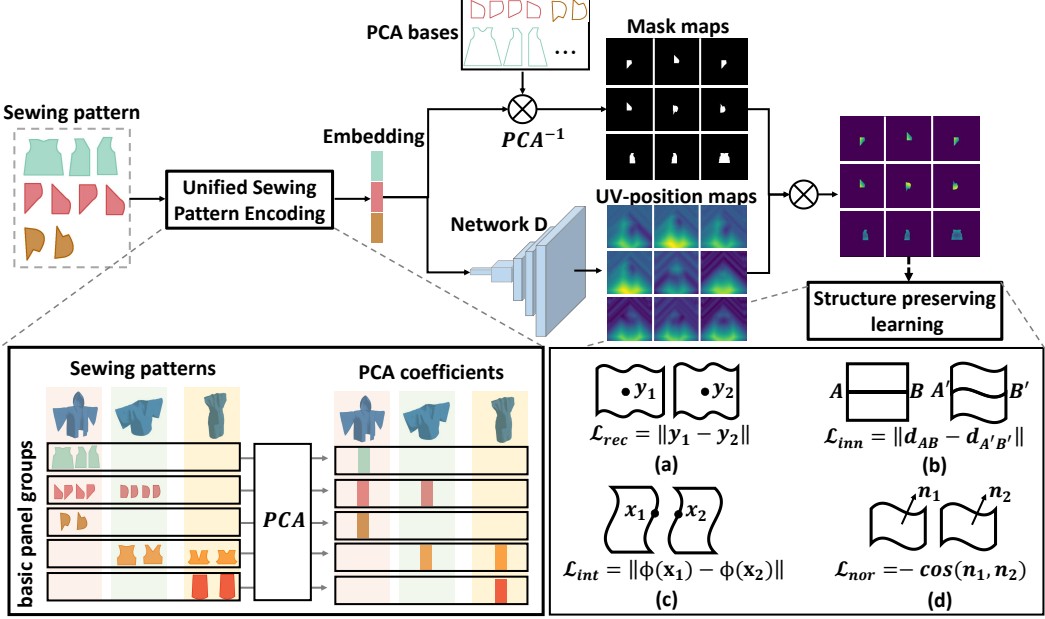

Figure 4: This is the Framework of our NSM. The sewing pattern is first encoded into an embedding by the unified sewing pattern encoding module. Then, the UV-position maps with masks are predicted from the embedding by a neural network $D$ and the inverse PCA transform $PCA^{-1}$. We train NSM by optimizing four losses to ensure 3D garment structure preservation: (a) reconstruction loss; (b) inner-panel structure-preserving loss; (c) inter-panel structure-preserving loss; (d) surface-normal loss.

the dilemma of not being able to model generic clothes. Even state-of-the-art methods still rely on physical simulators for clothing generation [16], failing to learn reconstructable and manipulable representations. Unlike all previous template-based methods, our NSM can model garments with complicated topologies; Novelly, our NSM uses a sewing machine to solve the generic garment modeling problem that is intractable with existing template-based methods.

**Learned UV parameterization.** UV parameterization is the process of mapping a 3D surface to a 2D plane. Usually, this mapping is bijective and means that we can store the 3D vertices continuously at a 2D UV plane. This allows 2D convolutional neural networks to operate on 3D shape, which has been introduced into neural garment modeling called UV-position maps [28, 30, 41, 45, 46, 17, 18]. To formulate the 3D garment deformation, some works [45, 46, 17, 41, 18] introduce the UV-position maps for enhancing and producing realistic and detailed garment surface deformation from coarse mesh parameterization. Other works [28, 30] use UV-position maps for encoding 3D garment with different shapes and topologies and utilize the auto-encoder structure to learn garment representation or generation. However, existing methods use either fixed pre-defined UV maps or rely on human body mesh parameterization. In contrast, our NSM makes flexible use of UV-position maps with the help of sewing patterns. We observe that the sewing pattern is one solution of the UV mapping process with low distortion as garment is simulated by sewing the panels on garment surface. Hence, we simultaneously model the 2D panel shapes and garment 3D shapes on UV-position maps, allowing our NSM to represent garments with diverse shapes and topologies.

## 3 Neural Sewing Machines

In this work, we study the problem of modeling 3D garments with diverse shapes and topologies. Previous methods [30, 24, 32, 21] were constrained to model garments under specific categories or with relative simple topologies, and failed to learn reconstructable and manipulable representations. To this end, we propose Neural Sewing Machine (NSM), a learning framework for structure-preserving 3D garment modeling capable of learning representations for garments with diverse shapes and topologies, as shown in Fig. 4. As sewing patterns accurately describe the intrinsic structure and the topology of the 3D garment, we first use a unified garment encoding module to obtain an embedding from the sewing pattern. Then, we use a 3D garment decoding module to decode sewing pattern embeddings to 3D garment shape using the representation of UV-position maps with masks [30]. To allow 3D structure-preserving learning, we introduce an inner-panel loss, an inter-panel loss, and a surface-normal loss. In the following, we will demonstrate the components of NSM in detail.

## 3.1 Unified Sewing Pattern Encoding

We first introduce the definition of the sewing pattern, then we demonstrate how to encode the sewing patterns from different garment categories into a shared latent space.

Previously, [15] defined a sewing pattern by a set of 2D panels along with the stitch information about how the panels are stitched together. For example, in Fig. 3, a sewing pattern of a skirt had 4 panels named *top front*, *top back*, *dress front* and *dress back*. Each panel corresponded to a part of the 3D garment, and the correspondences were featured by the colors. The stitch information of how panels were stitched together on the 3D garment shape was featured by the arrowed dashed lines between panel edges.

To obtain unified sew pattern embeddings for generic garments with different categories and shapes, we extend the definition in [15] to a hierarchical graph (see Fig. 5). Formally, a garment is defined by a sewing pattern, and a sewing pattern is defined by a graph $G = (V, E)$, where the graph node $V$ represents the set of all seamed edges of all panels in a garment; the graph adjacencies $E$ represents the stitch information of how panels were stitched together. Then, the previous definition of a sewing pattern can be rewritten by $G = (P(V), E)$, where $P(V)$ is a panel containing several seamed edges. However, using $(V, E)$ or $(P, E)$ to describe a garment could be too specific and will overfit a garment, which is unsuitable for describing generic garments and for getting generic sewing pattern embeddings. To obtain generic sewing pattern embeddings, we introduce basic panel groups. As shown in Fig. 5, a basic panel group $B(V)$ or $B$ is a combination of several panels, and thus a garment (i.e., a sewing pattern) can be represented as a combination of the basic panel groups $B(V)$. For example, the skirt in Fig. 5 contains two basic panel groups; the three garments in the left concern of Fig. 4 (each column is a garment) contain three, two, two basic panel groups, respectively. Finally, a sewing pattern is rewritten by $G = (B(V), E)$. Note that the basic panel groups are statistically collected from all garment categories in the sewing patterns dataset [15].

With the definition of sewing patterns in terms of basic panel groups, we are able to compute unified sewing pattern embeddings for a garment. **First**, we discrete each seamed edge into $m$ points, with each point represented by a 2D coordinate, and for a given basic panel group $B_i$ having $l$ seamed edges, we have a raw representation $B_i \in \mathbb{R}^{l \times m \times 2}$. **Second**, assume that there are a total of $N$ data containing this basic panel group $B_i$ in the whole dataset; then, we compute the mean shape $\bar{B}_i \in \mathbb{R}^{l \times m \times 2}$ for these $N$ samples. **Third**, we compute a PCA subspace for these $N$ samples after subtracting the mean shape $\bar{B}_i$ and we obtain the PCA coefficients

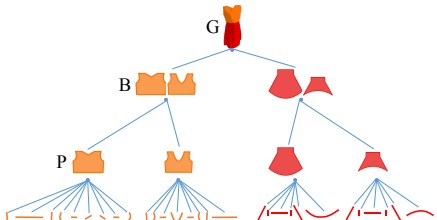

Figure 5: Definition of a sewing pattern in terms of basic panel groups.

$\gamma_i^n \in \mathbb{R}^h$ ($1 \leq n \leq N$) for each sample and PCA components $A_i \in \mathbb{R}^{l \times m \times 2 \times h}$, where $h$ is the number of PCA components. **Fourth**, an embedding $\gamma$ of a sewing pattern is obtained by concatenating all the PCA coefficients $\gamma_i$ of the basic panel groups (leaving blank if a garment or a sewing pattern does not contain some basic panel groups). Fig 4 presents a detailed illustration of these processes.

## 3.2 3D Garment Decoding

Given the embedding $\gamma$ for a sewing pattern sample, we decode the 3D garment shape that is draped on the T-pose human body. To represent 3D garment under diverse categories and encrypt the topologies and shape details of the garments, UV-position maps with mask maps [28, 30] are introduced to represent the 3D garment shape, which store the 3D coordinates of the garment at its UV coordinates and the masks indicate the 2D panels shapes of the 3D garment, as shown in Fig. 4. Note that previous methods only use a single map to store 3D garment by registering the 3D garment onto human body, which has limitation for representing loose garment or garment with complex topology. In contrast, we introduce multiple maps with each map individually representing one panel of the garment, which is capable of representing garment with arbitrary topology and shape.

Specifically, we use a CNN-based network $D$ to predict the UV-position maps for panels $\{Y^t\}_{t=1}^T$ from the garment representation $\gamma$, where $Y^t \in \mathbb{R}^{H \times W \times 3}$ is the UV-position map for the panel and $T$ is the panel number. The network $D$ paraterized by $\theta_D$ uses the Decoder architecture in [27]:

$$\{Y^t\}_{t=1}^T = D(\gamma \, ; \, \theta_D). \tag{1}$$

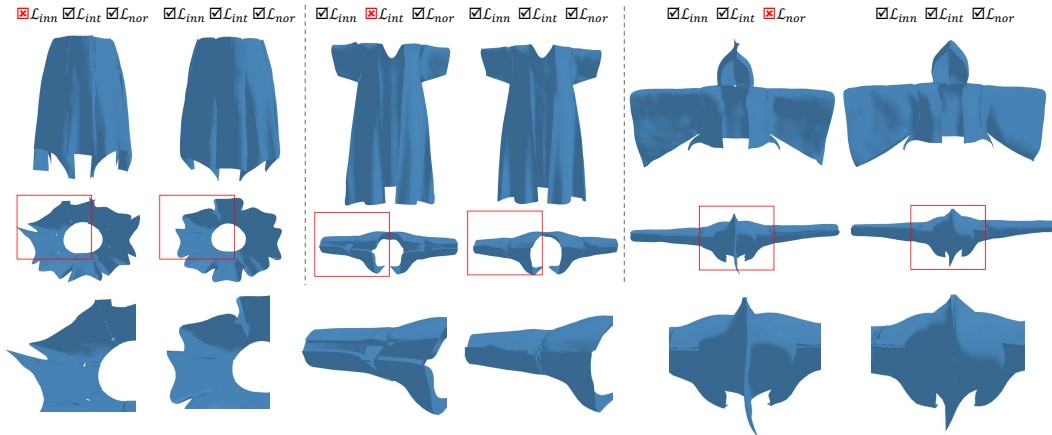

Figure 6: Illustration of the losses in structure preserving learning. $\mathcal{L}_{inn}$ is benefit for learning reasonable inner-panel deformation. $\mathcal{L}_{int}$ helps to preserve the inter-panel topology structure. $\mathcal{L}_{nor}$ helps to preserve local structure on the garment.

The mask maps are introduced to represent the detailed 2D panel shape of the garment. Instead of predicting the mask maps by a neural network, which tends to miss the small parts of the masks. We use the inverse PCA transform $PCA^{-1}$ to decode the mask maps $\{M^t\}_{t=1}^{T}$ from the garment representation $\gamma$, where $M^t \in \{0,1\}^{H \times M}$ indicate the 2D panel shape:

$$\{M^t\}_{t=1}^{T} = PCA^{-1}(\gamma). \tag{2}$$

The 3D garment mesh can be readout from the UV-position maps $\{Y^t\}_{t=1}^{T}$ and the mask maps $\{M^t\}_{t=1}^{T}$ in 3D space. The garment readout process is provided in the Appendix.

## 3.3 Structure Preserving Learning

It is not trivial to learn the structured deformable 3D garment with diverse shapes and topologies from the embedding that is encoded from the 2D sewing patterns. To preserve the intrinsic structure of the predicted 3D garment, except from the 3D reconstruction loss, we introduce an inner-panel structure-preserving loss, an inter-panel structure-preserving loss, and a surface-normal loss in the learning process of our NSM. In the following, we introduce the training losses in our NSM.

**3D Reconstruction Loss:** We optimize the predicted UV-position maps $\{Y^t\}_{t=1}^{T}$ using the 3D reconstruction loss $\mathcal{L}_{rec}$ as following, where $\hat{Y}^t$ is the ground truth UV-position map and $\hat{M}^t$ is the ground truth mask map for the panel. The loss $\mathcal{L}_{rec}$ constrains the 3D coordinates of the prediction and the ground truth should be the same at the corresponding UV-coordinates:

$$\mathcal{L}_{rec} = \sum_{t=1}^{T} \left\| Y^t * \hat{M}^t - \hat{Y}^t \right\|_{1}. \tag{3}$$

**Inner-panel Structure Preserving Loss:** In practice, the neural networks tend to learn the low frequency signal [24] and a garment usually has folds on its surface. The 3D Reconstruction loss solely cannot guarantee the network $D$ to learn an isometric mapping between the UV-coordinates and the 3D coordinates, as shown in Fig. 6. The garment surface is flat as the panels are deformed with a wrong scale. Hence, we introduce the inner-panel structure preserving loss $\mathcal{L}_{inn}$ to learn an isometric mapping, which helps to preserve the intrinsic structure within each deformed panel, where $s$ is a constant that indicates the scaling between the 3D coordinate and the UV coordinate. The loss $\mathcal{L}_{inn}$ constrains that the length of a curve on a panel should be consistent no matter how the panel is deformed in 3D space:

$$\mathcal{L}_{inn} = \sum_{t=1}^{T} \sum_{i=1}^{H-1} \sum_{j=1}^{W-1} \left\| \left\| Y_{i,j}^t - Y_{i+1,j}^t \right\|_2 - s \right\|_1 + \left\| \left\| Y_{i,j}^t - Y_{i,j+1}^t \right\|_2 - s \right\|_1. \tag{4}$$

**Inter-panel Structure Preserving Loss:** The above losses only model the inner-panel geometry and topology. As a garment consists of different panels that are stitched together following the stitch information, we use the inter-panel structure preserving loss $\mathcal{L}_{int}$ to encode this inter-panel topology knowledge. As shwon in Fig. 6, when the loss $\mathcal{L}_{int}$ is removed, there exists a gap between two edges that should be stitched together. Specifically, given two stitched edges $e_{l_1}^{t_1}$ and $e_{l_2}^{t_2}$ on the panels

$P^{t_1}$ and $P^{t_2}$ using the stitch information $E$, illustrated in Figure 5, we use bilinear interpolation $\Phi$ to obtain their 3D coordinates on their UV-position maps $Y^{t_1}$ and $Y^{t_2}$, where $e_{l_1}^{t_1} \in \mathbb{R}^{m \times 2}$ and $e_{l_1}^{t_1} \in \mathbb{R}^{m \times 2}$ denote the UV coordinates of $m$ points on the two edges. The loss $\mathcal{L}_{int}$ demonstrates that two stitched edges are point-wise stitched together:

$$\mathcal{L}_{int} = \sum_{(e_{l_1}^{t_1}, e_{l_2}^{t_2}) \in S} \left\| \Phi(e_{l_1}^{t_1}, Y^{t_1}) - \Phi(e_{l_2}^{t_2}, Y^{t_2}) \right\|_1. \tag{5}$$

**Surface Normal Loss:** To further approximate the deformation on the 3D garment surface, we introduce the surface norm loss $\mathcal{L}_{nor}$ as following. As shown in Fig. 6, the loss $\mathcal{L}_{nor}$ helps to preserve local structure on the garment surface. The normal map $N^t$ is first calculated from the predicted UV-position map $Y^t$ by finding the normal vector that is perpendicular to the tangential plane of each 3D point from the UV-position map $Y^t$. Then we calculate the *cosin* loss for each point on the 3D garment surface as following with ground truth normal map $\hat{N}^t$ and the mask map $\hat{M}^t$:

$$N^t = -\frac{\partial Y^t}{\partial u} \times \frac{\partial Y^t}{\partial v} / (\|\frac{\partial Y^t}{\partial u}\| \|\frac{\partial Y^t}{\partial v}\|). \tag{6}$$

$$\mathcal{L}_{nor} = -\sum_{t=1}^{T} \sum_{i=1}^{W} \sum_{j=1}^{H} cos(N_{i,j}^t * \hat{M}_{i,j}^t, \hat{N}_{i,j}^t). \tag{7}$$

The total training loss $\mathcal{L}$ in the learning process of NSM is given as following:

$$\mathcal{L} = \alpha_{rec}\mathcal{L}_{rec} + \alpha_{inn}\mathcal{L}_{inn} + \alpha_{int}\mathcal{L}_{int} + \alpha_{nor}\mathcal{L}_{nor}. \tag{8}$$

# 4   Experiments

**Dataset.** Here we use the largest 3D garment dataset with sewing patterns introduced in [15]. It covers a variety of garment designs, including variations of t-shirts, jackets, dresses, skirts, jumpsuits, and pants. Compared with the previous 3D garment datasets [32, 4], this dataset contains more diverse 3D garment shapes and topologies. We use about 22400 samples from 12 base categories in the experiments; each sample contains a sewing pattern, 3D garment mesh draped on T-pose SMPL [20] and a rendered image of the 3D garment. We use the first 90% part of the data for each base type as the training set and the remaining as the test set.

**Evaluation Metrics.** To evaluate the quality of 3D garment reconstruction, following [47, 38], we calculate the Chamfer distance (Chamfer) and the average point-to-surface Euclidean distance (P2S) between the predicted and ground truth 3D point clouds. Furthermore, to evaluate the quality of the topology structure of the 3D garment, we introduce the mean geodesic length error (MGLE) to measure the topology structure difference between two 3D meshes. Given two points $\boldsymbol{x}_i$ and $\boldsymbol{x}_j$ on the ground truth 3D mesh surface, we find two points $\boldsymbol{x}_i'$ and $\boldsymbol{x}_j'$ on the predicted 3D mesh surface that are closet to points $\boldsymbol{x}_i$ and $\boldsymbol{x}_j$ respectively. The geodesic length error is calculated as follows:

$$MGLE = \frac{2}{K(K-1)} \sum_{i=1}^{K-1} \sum_{j=i+1}^{K} \left\| g(\boldsymbol{x}_i, \boldsymbol{x}_j) - g(\boldsymbol{x}_i', \boldsymbol{x}_j') \right\|_1, \tag{9}$$

where $g$ is the geodesic length between two points on the mesh computed using the algorithm in [9]. For each garment, as the calculation of geodesic length is relatively slow, we sample $K = 20$ points and calculate MGLE between every two points. The sensitivity analysis of $K$ is provided in the Appendix.

**Implementation Details.** The weights $\{\alpha_{rec}, \alpha_{inn}, \alpha_{int}, \alpha_{nom}\}$ in the training loss $\mathcal{L}$ are set to $\{1, 10^{-3}, 10^{-4}, 10^{-2}\}$. The number of PCA components $h$ is set to 12, which guarantees the sewing pattern shape is reconstructed with small errors. The basic panel group number are set to 10 and the total panel number is set to 33. The scaling $s$ between the 3D coordinate and the UV coordinate is set to 1.5. The width $W$ and height $H$ of the UV-position maps and the mask maps are set as 128. Our model is trained with a GTX 2080 GPU with the learning rate as $1e^{-3}$ and the batch size as 8 for 40 epochs.

## 4.1 Comparison of Single-view Reconstruction

To validate the generalization ability of our framework, we conduct experiments on single-view 3D shape reconstruction. We first train NSM on the training set with paired sewing patterns and the 3D garment annotations. Then we train an encoder for mapping the input image to the sewing pattern embedding $\gamma$ using a CNN-based architecture [27] as the backbone of the encoder. More details about training process are provided in Appendix. [38], AnchorUDF [47] and BCNet [13]. The results are shown in Table 1. We compare our NSM with three state-of-the-art single-view reconstruction methods, i.e., Pixel2Mesh Pixel2Mesh represents 3D objects as deformable 3D meshes, and AnchorUDF uses Unsigned Distance Function (UDF) to represent 3D garments, and BCNet uses a template mesh to represent garments for each cate-

Table 1: Chamfer, P2S errors and MGLE (cm) for different single-view reconstruction methods on the 3D garment dataset.

| Methods | Chamfer ↓ | P2S ↓ | MGLE ↓ |
|---|---|---|---|
| Pixel2Mesh [38] | 5.23 | 3.84 | 10.81 |
| AnchorUDF [47] | 3.31 | 4.17 | 4.61 |
| BCNet [13] | 4.69 | 4.33 | **3.42** |
| NSM | **2.08** | **1.90** | 3.73 |

gory. Our NSM achieves a 2.08cm error for chamfer, a 1.90cm error for P2S, and a 3.73cm error for MGLE, achieving better performance than all methods for Chamfer and P2S, achieving better performance than Pixel2Mesh and AnchorUDF for MGLE. Note that BCNet uses template meshes with predefined topology and our method achieves compatible performance with BCNet for MGLE.

**Visually**, as the three competitors implicitly model 3D garment shapes from compressed features or use registered template mesh, they may have limitations for recovering detailed garment structures, as shown in Fig. 7. In contrast, our NSM explicitly models the garment structure by introducing the sewing patterns, which helps to preserve the detailed garment structures. Besides, these three methods either use topology templates or recover the topology via post-process, which brings errors in modeling complicatedly-structured garments (e.g., wrinkled clothes and fine structures), as shown in Fig. 7. Differently, our NSM can recover detailed garment topology structures thanks to the learnable UV-position maps with masks.

## 4.2 Reconstruction from Sewing Pattern

To validate the garment representation ability of our NSM, we evaluate NSM for reconstructing 3D garments from sewing patterns. We train NSM on the training set with sewing patterns and 3D garment annotations. In the testing, given a sewing pattern, we first obtain the embedding by PCA transform using the PCA components pre-computed at the training phase. Then, the UV-position maps with masks (= 3D garments) are predicted from the embedding using the 3D garment decoder.

We report the experimental results in Table 2; the Chamfer distance error is 1.65cm, the P2S error is 1.46cm, and the MGLE error is 3.54cm. Fig. 1 presents the qualitative visualization, showing that our framework can effectively represent 3D garments from different

Table 2: Chamfer, P2S errors and MGLE (cm) obtained by our framework for 3D garment reconstruction from sewing pattern on the 3D garment dataset.

| NSM | t-shirt | jacket | dress | skirt | jumpsuit | pants | All |
|---|---|---|---|---|---|---|---|
| Chamfer ↓ | 1.48 | 2.17 | 1.77 | 1.61 | 0.66 | 1.53 | 1.65 |
| P2S ↓ | 1.17 | 2.08 | 1.27 | 1.19 | 0.73 | 2.13 | 1.46 |
| MGLE ↓ | 3.13 | 4.13 | 4.26 | 3.46 | 2.32 | 2.65 | 3.54 |

categories with diverse shapes and topologies. More comparisons are provided in the Appendix.

## 4.3 Controllable Garment Editing

We demonstrate our framework is capable of controllable garment editing. Given a source 3D garment and the predicted sewing pattern (obtained using a PointNet [26] trained on paired 3D garments and sewing pattern embeddings), we can accurately edit the 2D panel shapes or classes, from which our NSM recovers the corresponding edited 3D garment. Previous methods [28, 30] model and edit 3D garments via the implicit neural representation, having difficulty with controllable garment editing. In contrast, sewing pattern embedding is more interpretable, and the structure-preserving 3D garment modeling guarantees that the editing on 2D panels can be accurately reflected on 3D garments.

As shown in Fig. 8, NSM supports freely editing on the 3D garment shape and topology with preserved intrinsic structure. And NSM can edit garments with significant shape variations or transfer

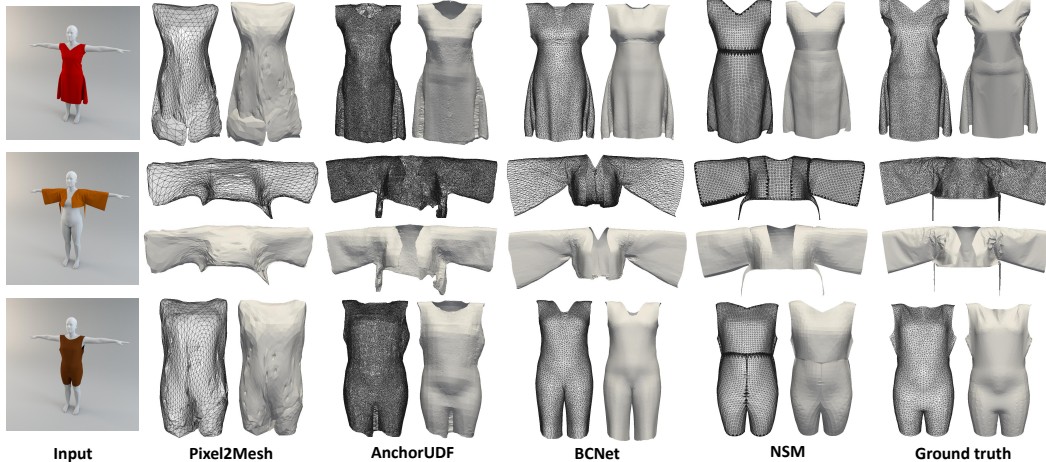

| Input | Pixel2Mesh | AnchorUDF | BCNet | NSM | Ground truth |

Figure 7: Visual comparison of the learned topology and geometry of different single-view reconstruction methods on the 3D garment dataset.

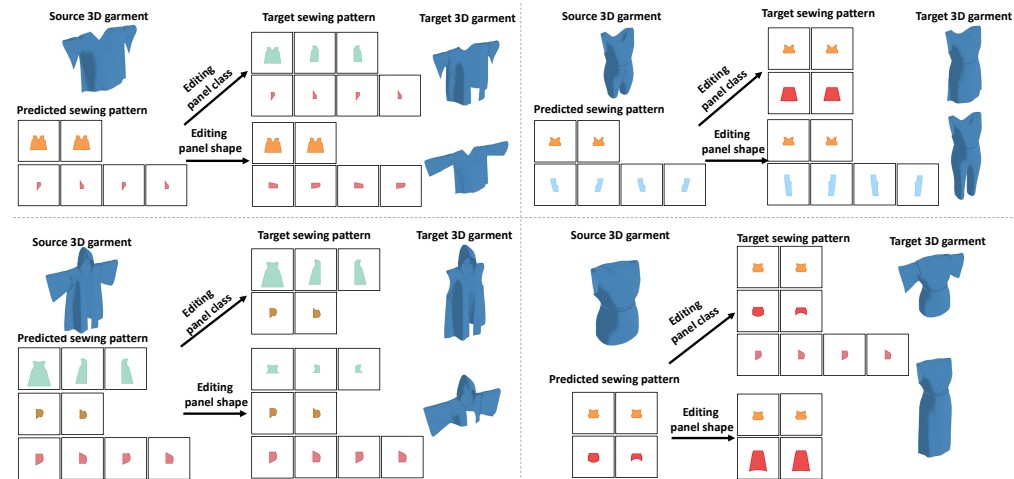

Figure 8: Illustration of controllable garment editing. Given a source 3D garment and the predicted sewing pattern, we obtain the edited target garment by editing the panel shape or panel class of the sewing pattern.

garments from one category to another by editing on the 2D panels, even for some categories that are not shown in the training set.

## 4.4 Model Analysis

**Ablation Study.** We conduct ablation studies to provide insights into the effectiveness of each component in our framework, as shown in Table 3.

Only using the 3D reconstruction loss $\mathcal{L}_{rec}$ for training, the framework achieves an MGLE error of 4.38cm. When the inner-panel structure-preserving loss $\mathcal{L}_{inn}$ is added, the MGLE error increases from 4.38cm to 4.69. This might be because inner-panel structure-preserving loss $\mathcal{L}_{inn}$ affects the inter-panel topology structure. When the inter-panel structure-preserving loss $\mathcal{L}_{int}$ is added, the MGLE error drops from 4.69cm to 3.62. When the surface normal loss $\mathcal{L}_{nor}$ is added, our full model achieves the MGLE error as 3.54cm. Note that the changes in Chamfer and P2S error are not obviously in our ablation study. But as shown in Figure 6, when the losses used for structure-preserving learning are removed, the model can not output 3D garments with correct structures. This indicates that chamfer and P2S errors cannot truly reflect the quality of the topology structure of the 3D object. On the contrary, the MGLE error can reflect the topology structure of the 3D object since it evaluates the geodesic length error on the 3D object surface.

Table 3: Quantitative ablation study.

| $\mathcal{L}_{rec}$ | $\mathcal{L}_{inn}$ | $\mathcal{L}_{int}$ | $\mathcal{L}_{nor}$ | Chamfer ↓ | P2S ↓ | MGLE ↓ |
|---|---|---|---|---|---|---|
| ✓ | | | | 1.81 | 1.69 | 4.38 |
| ✓ | ✓ | | | 1.80 | 1.76 | 4.69 |
| ✓ | ✓ | ✓ | | 1.80 | 1.68 | 3.62 |
| ✓ | ✓ | ✓ | ✓ | **1.65** | **1.46** | **3.54** |

**Interpolation Results.** We also show the results of our NSM for interpolating in the sewing pattern embedding space, as shown in Fig. 9. Given the a source embedding $\gamma_s$ and a target embedding $\gamma_t$,

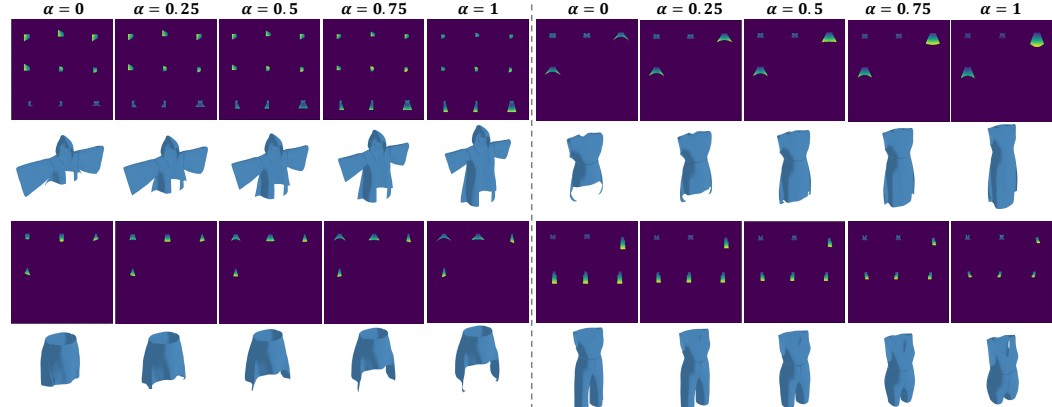

Figure 9: Visualization of the interpolation in the sewing pattern embedding space of NSM. The first and third rows denote the predicted UV-position maps with masks. The second and forth rows denote the predicted 3D garment shapes.

the interpolated embedding is given as $\gamma_i = \alpha\gamma_s + (1-\alpha)\gamma_t$. We predict the UV-position maps with masks (= 3D garments) from the interpolated embedding $\gamma_i$. As we can see, the changes in 2D panels and 3D garment shapes are consistent.

## 5   Conclusion

We present a Neural Sewing Machine (NSM) for structure-preserving 3D garment modeling. The sewing pattern embedding is obtained by the unified sewing pattern encoder. Then the UV-position maps with masks are predicted from the embedding. To preserve the intrinsic structure of the 3D garment, we introduce inner-panel structure loss, inter-panel structure loss, and surface-normal loss. Experiments show that NSM is capable of representing garments with diverse shapes and topologies, reconstructing 3D garments from single images, and editing 3D garments by controlling the panel shape and class.

**Acknowledgement.** This work was supported in part by National Key R&D Program of China under Grant No.2020AAA0109700, National Natural Science Foundation of China (NSFC) under Grant No.U19A2073 and No.61976233, Guangdong Province Basic and Applied Basic Research (Regional Joint Fund-Key) Grant No.2019B1515120039, Guangdong Outstanding Youth Fund (Grant No. 2021B1515020061), the UKRI grant Turing AI Fellowship EP/W002981/1 and the EPSRC/MURI grant EP/N019474/1. We also thank the Royal Academy of Engineering.

## Limitations and broader impact

The method in this paper aims to build 3D garment models as realistic as possible, which has significant positive effects on the real world, such as facilitating garment design, virtual try-on, and virtual human design in the metaverse. Currently, we do not model the garment deformation with different human pose. This is a challenge problem for loose garments with diverse shape variations. Our solution is to use "virtual garment bones" [23] or diffused skinning [19]. Another limitation of our work is that our NSM encoder requires the panels to be relatively regular. While some garments may have irregular panels, like the several garments found in South Asia, East Asia and Africa. We will expand this in our future work. We believe that this work does not have any negative impact on society.

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
