# OpenReview forum: "Structure-Preserving 3D Garment Modeling with Neural Sewing Machines"
_NeurIPS.cc/2022/Conference — NeurIPS 2022 Accept_

### Official Review · Reviewer_WXEe · 2022-07-08

**Rating:** 5
**Confidence:** 4
**Soundness:** 3 good
**Presentation:** 3 good
**Contribution:** 3 good

**Summary:**

This paper proposes the Neural Sewing Machine (NSM) for structure-preserving 3D garment modeling. Previous methods mostly model garments in a class-specific way, which is not flexible and easily extendable. In this work, each garment is modeled as several basic panels and their stitching relations. The basic panels are encoded by PCA from a large-scale dataset. The stitching relations as well as the 3D structure of each panel are encoded in the UV-position maps. For each image input, NSM predict PCA coefficients for each basic panel that corresponds to the observed garment. Based on the predicted PCA coefficient, a CNN-based network is trained to predict the UV-position maps. Four training targets are introduced to facilitate the supervised training. The reconstruction results are the best compared with previous baselines. Utilizing the structure-preserving modeling, controllable garment editing and interpolation are also shown to demonstrate the versatility of this kind of garment modeling.

**Questions:**

1. Section 4.2, what is the input for the reconstruction from sewing pattern?
2. Why inner-panel structure preserving loss learns an isometric mapping between UV and 3D coordinates?
3. For the controllable garment editing, can we draw the panel shape by hand?

**Limitations:**

The limitation is not discussed. Please discuss more on the limitation. Please see the above 'Weakness' section for my comments on the proposed garment modeling.

**Strengths And Weaknesses:**

## Strength
1. Considering the sewing pattern into the garment modeling process and unifying modeling for basic garment classes are important contributions towards practical real-world garment modeling.
2. Both qualitative and quantitative results surpasses previous SOTA methods.
3. Ablation study proves the effectiveness of the proposed training targets.
4. The application of controllable garment editing is interesting.

## Weaknesses
1. The experiments are all done on a synthetic dataset, which might not be enough to fully prove its practicality when facing real-world data. For example, given a real photo of a clothed human, can NSM still give correct reconstructions on the clothes?
2. It seems like the sewing pattern/ topology of several basic panels are solely represented by the predicted UV-position maps. It might lead to two problems. a) For two corresponding edges, the predicted UV-position are not guaranteed to be perfectly aligned, which would give broken geometry when stitching panels. b) For problem a, a solution would be to find correspondences and force them to align spatially. However, finding such correspondences is not trivial under the current modeling. The edge points in the predicted UV-position map are not necessarily spatially ordinal.
3. It seems like NSM is only capable of reconstruct clothes in the canonical space. Can NSM takes photos of a posed clothed human as input? Can NSM output posed clothes as observed in the input image? A simple synthetic dataset for this would be Cloth3D.

## Typo
Table 3, headers, four L_{rec}.

## Missing References
[1] Xiang, D., Prada, F., Wu, C., & Hodgins, J. (2020, November). Monoclothcap: Towards temporally coherent clothing capture from monocular rgb video. In 2020 International Conference on 3D Vision (3DV) (pp. 322-332). IEEE.

[2] Hong, F., Pan, L., Cai, Z., & Liu, Z. (2021). Garment4D: Garment reconstruction from point cloud sequences. Advances in Neural Information Processing Systems, 34, 27940-27951.

[3] Bhatnagar, B. L., Sminchisescu, C., Theobalt, C., & Pons-Moll, G. (2020, August). Combining implicit function learning and parametric models for 3d human reconstruction. In European Conference on Computer Vision (pp. 311-329). Springer, Cham.

[4] Alldieck, T., Magnor, M., Bhatnagar, B. L., Theobalt, C., & Pons-Moll, G. (2019). Learning to reconstruct people in clothing from a single RGB camera. In Proceedings of the IEEE/CVF Conference on Computer Vision and Pattern Recognition (pp. 1175-1186).

---

> ### Author Response · Authors · 2022-08-02
> ****Response to Review WXEe** (Part 2)**
>
>
>
> > Concern 4: In Section 4.2, what is the input for the reconstruction of the sewing pattern?
>
> **A4:** Sorry for the confusion! In Section 4.2, the input is a sewing pattern, and the output is a predicted 3D garment mesh. Specifically, given a sewing pattern at the inference stage, we first use the PCA transform obtained at the training stage to map it into a latent embedding. Then it is fed into our NSM garment decoder to predict the UV-position maps, from which a 3D garment mesh is a read-out. We evaluate the reconstruction error between the predicted 3D garment mesh and the ground truth 3D garment mesh.
>
> > Concern 5: Why inner-panel structure preserving loss learns an isometric mapping between UV and 3D coordinates?
>
> **A5:** Thank you so much for your careful reading! The inner-panel structure preserving loss constrains that the distance between two near points on UV should be maintained after transformation in 3D only except for a constant ratio, as the measure of length is not exactly the same between the UV maps and the 3D shape. This loss approximately learns an isometric mapping when two points are sampled very near to each other since a 3D garment can be regarded as a 2D manifold structure in 3D and "locally" resembles Euclidean space.
>
> > Concern 6: For the controllable garment editing, can we draw the panel shape by hand?
>
> **A6:** Many thanks! Garment editing can be **either automatic or manual**. If you like manual drawing, of course, you can draw it. Otherwise, garment editing can be realized by designing a GUI for simply dragging the positions of the vertices or changing the curvature between two vertices on the panels. Thereafter, our model will quickly return the edited 3D garment.
>
> > Concern 7: The limitation is not discussed. Please discuss more on the limitation. Please see the above 'Weakness' section for my comments on the proposed garment modeling.
>
> **A7:** Thanks! We will discuss more limitations (especially your concerns) in the revision.
>
> > Concern 8: Missing References.
>
> **A8:** Thank you so much for recommending us these important related works! We have updated the missing references in our revision.
>
>
> [1] Kocabas et al. "Vibe: Video inference for human body pose and shape estimation.", 2020.
>
> [2] Gong et al. "Graphonomy: Universal human parsing via graph transfer learning.", 2019.
>
> [3] Pan et al. "Predicting Loose-Fitting Garment Deformations Using Bone-Driven Motion Networks.", 2022

---

> ### Author Response · Authors · 2022-08-02
> ****Response to Review WXEe** (Part 1)**
>
>
> We thank reviewer WXEe for the thoughtful comments. We are encouraged that the reviewer found our approach valuable to the community. Please see our responses below.
>
> > Concern 1: The experiments are all done on a synthetic dataset, which might not be enough to fully prove its practicality when facing real-world data.
>
>
>
> **A1:** Thanks a lot! As advised, **we used the real-captured in-the-wild images for evaluation**. Given an image of a T-pose person, we first estimate the camera parameters using [1] and a 2D cloth semantic segmentation using [2], then we fit the trained NSM to the image to obtain the 3D garment. We set the input embedding for the NSM decoder as learnable variables and fixed the NSM decoder parameters. We optimized the projection of the predicted garment to match the cloth segmentation on the image. The results will come soon these days because we spent lots of time running the experiments posed by Reviewer HS7n in the past week. (We have updated the visualization results in Sec5 of our revised supplementary)
>
>
>
> > Concern 2: The sewing patterns/topologies of several basic panels are solely represented by the predicted UV-position maps. It might lead to two problems. a) For two corresponding edges, the predicted UV position is not guaranteed to be perfectly aligned, which would give broken geometry when stitching panels. b) For problem a, a solution would be to find correspondences and force them to align spatially.
>
> **A2:** Thanks for your concern on this issue!
> We appreciate your interest in our implemental details! Actually, the sewing patterns/topologies of several basic panels are **NOT** solely represented by the predicted UV-position maps. Instead, they should be stitched together.
>
>
>  We first clarify several concepts as follows.
>
> **Sewing pattern:** A sewing pattern consists of the stitch information for which two edges from different panels should be stitched together.
>
> **Raw output of NSM:** The raw output of our NSM consists of the UV-position map and the panel contour for each panel.
>
>
> With these concepts, we can easily describe how the position maps compose the final 3D mesh.
>
> **Step 1: Construct the triangulated mesh for each panel.** The grid points on the UV-position maps are chosen as the inner vertices if they are inside the panel contour. Then we sample the points on the panel contour as the edge vertices. We uniformly sample a fixed number of vertices on each contour edge. Then we construct the triangulated mesh from these vertices on a 2D plane by an automatic triangulation algorithm. The 3D coordinate of each vertex is obtained by bilinear interpolation on the UV-position map.
>
> **Step 2: Construct the inter-panel triangulated mesh.**
> As the stitching information is known, we first determine which two edges from panels are stitched. In general, the panels are stitched uniformly along the edge as we have uniformly sampled a fixed number of vertices on each edge. We also apply the automatic triangulation algorithm to obtain the inter-panel triangulated mesh.
> As we have introduced the inter-panel structure preserving loss, the deviation of two edges on the most predicted 3D garment is relatively small. The results without post-processing are shown in Figure 7 in the main paper, Figure 3, and the demo video in the supplementary.
>
> **Step 3: Post-processing (optional).** Post-processing can also be introduced to further prevent the artifacts using Laplacian mesh editing or physical-based simulation on MAYA. (see A3 to Reviewer 1Ro2)
>
>
>
> > Concern 3: It seems like NSM is only capable of reconstructing clothes in the canonical space. A simple synthetic dataset for this would be Cloth3D.
>
> **A3:** Thanks for this imaginative question! Your mentioned Cloth3D dataset doesn't provide the sewing pattern annotations, disabling training our model on the Cloth3D dataset. The reason why we only reconstruct garments with the canonical pose is attributed to benchmark availability, i.e., existing sewing-pattern benchmarks only contain canonical-pose data. We will expand NSM to multi-pose bodies in our future work, dependent on the benchmark availability. One potential solution is to use "virtual bones" techniques (see [3]) for modeling loose garment deformation, which is beyond the scope of this paper.

---

### Official Review · Reviewer_RjUE · 2022-07-08

**Rating:** 6
**Confidence:** 4
**Soundness:** 2 fair
**Presentation:** 2 fair
**Contribution:** 2 fair

**Summary:**

This paper introduces a network model to reconstruct 3D garments based on sewing pattern input. The sewing patterns are first gone through PCA for coefficients to compose the embedding. Later, masks and uv position maps are generated using the coefficients. The pixels from the masked maps are used to reconstruct the 3D mesh. The paper introduces new loss functions to regularize the model behavior using deformation and sewing edge alignment rules, as well as normal consistentcy. Results show that the proposed method works better in a single-view garment reconstruction task and is capable of controllable editing and interpolation.

**Questions:**

 See above.

**Limitations:**

 See above.

**Strengths And Weaknesses:**

Strengths:
 - The paper is the first one that I know of that addresses the importance of modeling garments in the context of sewing patterns, which is the key to develop a physically correct as well as easy-to-use garment generation models.

Weakness:
 - The method does not model tightness of the cloth. What it really does is merely a *garment template mesh* generation model for a given sewing pattern shape. While it is not a critical component, the authors should discuss it either in future work or in the limitation section.
 - It is still not clear how the position maps compose the final 3D mesh. Although there are several sentences in the caption of the figure in the appendix that mentions it, it is not in sufficient detail. How is the mesh triangulated around the sewing pattern boundary where the edges are defined in lines but the inner vertices are defined as grid points? What does the algorithm do when the 3D coordinates of the sewing edges from two patterns are different? Is there any post-process to smooth out the neighboring vertex locations caused by the difference? If not, would that generate unnatural artifacts? There are a lot of critical details on this topic but the authors did not discuss them in detail.
 - The fabric material is not modeled. This is not a necessary component but should be mentioned in the future work section.
 - It is not clear how the basic panel groups are defined and it is not shown that what has been included would be sufficient to support *generic* garments. There seems to be only 9 basic panel groups. How are they chosen? Which factors are considered? Which garments can be supported by the 9 groups and which cannot? For example, can the model support T-shirts with pockets or dresses with lace/ribbon decorations using the 9 groups? It is a critical component of the paper but it is not discussed in depth.
 - It seems in Fig. 8 that the garment editing for the panel class is manual. It doesn't actually help too much because the user will have to specify the new pattern shape themselves which could be tedious.

---

> ### Author Response · Authors · 2022-08-02
> ****Response to Review RjUE** (Part 2)**
>
>
>
> > Concern 3: The fabric material is not modeled. This is not a necessary component but should be mentioned in the future work section
>
> **A3:** Thanks for this insightful and futuristic advice! We will add more discussions about the material modeling in the limitations and future work.
>
>
> > Concern 4: It is not clear how the basic panel groups are defined, and it is not shown that what has been included would be sufficient to support generic garments.
>
> **A4:** The basic panel groups are defined as the **shared primitives across the training set, with each primitive having a similar function and panel structure**. For example, the sleeve panels are shared between T-shirt and jacket, which can be considered as one basic panel group. Specifically, there exist 12 common garment categories in the used dataset, and we extracted ten groups, including one sleeve panel group, two top panel groups, one hood panel group, one pant panel group, one belt panel group, and four skirt panel groups. Ideally, the completeness of the basic panel groups can be appended constantly when novel garment categories are included. The pockets on T-shirts or the lace/ribbon decorations on dresses can also be considered as new basic panel groups and appended to the training set. When the training set is updated, normally, we only need to fine-tune the trained model on the training set, learning to attach the novel appended groups on garments.
>
> > Concern 5: It seems in Fig. 8 that garment editing for the panel class is manual. It doesn't actually help too much because the user will have to specify the new pattern shape themselves, which could be tedious.
>
> **A5:** Many thanks! Garment editing can be **either automatic or manual**. Specifically, garment editing can be realized by designing a GUI for simply dragging the positions of the vertices or changing the curvature between two vertices on the panels. Thereafter, our model will quickly return the edited 3D garment.

---

> > ### Comment · Reviewer_RjUE · 2022-08-08
> > **Updates**
> >
> > The authors have addressed my concerns thoroughly. After reading other reviews, I decided to change my score to 6. If accepted, please consider adding the details above into the appendix so that the paper is self-contained.

---

> ### Author Response · Authors · 2022-08-02
> ****Response to Review RjUE** (Part 1)**
>
>
>
> We are very grateful to Reviewer RjUE for recognizing the originality of this article and its importance to the community. Thank you so much for your insightful comments! We try to address all of your concerns and clarify some technical details. Please see our responses below.
>
> > Concern 1: The method does not model the tightness of the cloth.
>
> **A1:** Thanks for this visionary insight! Unfortunately, we are unable to model the tightness of the cloth due to dataset unavailability. One possible solution is to generate data where each garment sample has multiple sizes, such as small, medium, and large, and the garment can be draped on different body sizes, such as slim, medium, and large. However, this is beyond the scope of this paper and will be discussed as a limitation in our revision.
>
> Note that our NSM is promising in modeling garment tightness because it represents garments with sewing patterns, allowing us to represent garments with diverse shapes and topologies. This differs from template-based methods because they use a template mesh with a fixed vertex number and cannot represent garments with diverse shapes \& topologies to model tightness.
>
>
>
> > Concern 2: It is unclear how the position maps compose the final 3D mesh.
>
> **A2:** We appreciate your interest in our implemental details! We first clarify several concepts as follows.
>
> **Sewing pattern:** A sewing pattern consists of the stitch information for which two edges from different panels should be stitched together.
>
> **Raw output of NSM:** The raw output of our NSM consists of the UV-position map and the panel contour for each panel.
>
>
> With these concepts, we can easily describe how the position maps compose the final 3D mesh.
>
> **Step 1: Construct the triangulated mesh for each panel.** The grid points on the UV-position maps are chosen as the inner vertices if they are inside the panel contour. Then we sample the points on the panel contour as the edge vertices. We uniformly sample a fixed number of vertices on each contour edge. Then we construct the triangulated mesh from these vertices on a 2D plane by an automatic triangulation algorithm. The 3D coordinate of each vertex is obtained by bilinear interpolation on the UV-position map.
>
> **Step 2: Construct the inter-panel triangulated mesh.**
> As the stitching information is known, we first determine which two edges from panels are stitched. In general, the panels are stitched uniformly along the edge as we have uniformly sampled a fixed number of vertices on each edge. We also apply the automatic triangulation algorithm to obtain the inter-panel triangulated mesh.
> As we have introduced the inter-panel structure preserving loss, the deviation of two edges on the most predicted 3D garment is relatively small. The results without post-processing are shown in Figure 7 in the main paper and the demo video in the supplementary.
>
> **Step 3: Post-processing (optional).** Post-processing can also be introduced to further prevent the artifacts using Laplacian mesh editing or physical-based simulation on MAYA. (see A3 to Reviewer 1Ro2)

---

### Official Review · Reviewer_HS7n · 2022-07-09

**Rating:** 6
**Confidence:** 5
**Soundness:** 1 poor
**Presentation:** 2 fair
**Contribution:** 2 fair

**Summary:**

This paper looks into the problem of 3D garment modeling and reconstruction which is a challenging problem within the wider area of 3D generative modeling and reconstruction. The problem is especially challenging because it is unclear as to what constitutes a good representation for 3D clothes that captures the wide variety of garments we see around us. This paper proposes to use sewing patterns as a representation of 3D garments and proposes a learning based framework - Neural Sewing Machine - capable of generating garments of varied topology. The authors discuss challenges and propose solutions to both encoding and decoding using this representation. They additionally propose three kinds of losses – inner-panel structure preserving loss, inter-panel structure preserving loss and surface normal loss. The authors claim that their method outperforms the prior state of the art.


**Questions:**

1. L59 : “Our NSM guarantees …” – I don’t understand how this statement is true? It’s not that the 2D UV map is an isometric parameterization of any kind. Maybe I am missing something?

2. The authors make comments about physics-based simulators in several places and believe that their method is superior to this line of work. I have absolutely no clue how using physically based simulators for interacting with the clothes is a “contemporary line of work” to the proposed method, as the two are completely different items? What’s the context or argument here? Also, the authors claim that their method creates garments which can be manipulated but nowhere do they qualify or provide any evidence for this one.

3. In the dataset, the authors say L245 - “ We use the first 90% part of the data for each base type … and remaining as the test set”. Where is the validation set? If the authors fine tuned their model on the test set then this basically throws all of experimental evaluation into jeopardy.

4. I also didn’t understand the section on controllable garment editing. It looks interesting but I could not parse through the procedure involved in getting that.

I firmly believe that the idea has a lot of potential, but I am not convinced with the experimental evaluation provided here. Unless the authors can address the concerns (primarily with comparisons to relevant methods) I don't think there is any merit to accepting paper. This is simply because the main hypothesis of this paper is that using sewing patterns as a representation is inherently superior to other kinds of representation. And in the absence of state of the art results on relevant comparisons, I don't see how else the authors would validate this hypothesis.

**Limitations:**

I am surprised that the authors didn't talk about obvious limitations of their work — both technical and in terms of its social impact.

Due to the choice of their representation, I doubt that their method generalizes to other clothing styles and the authors neither experiment with the generalization nor provide any comments on it. This to me seems like a severe limitation of the proposed work. However, I don't think this is any reason to reject the paper or the idea as this is not the main hypothesis/claim of the paper.

The dataset used in this paper contains clothing styles which are not very inclusive. In particular, there are several garments found in South Asia, East Asia and Africa which would be hard to be captured by the PCA basis used in the paper — which primarily contains popular garments worn in the western countries. I believe that the lack of inclusivity of garments found in other (and often underrepresented) regions of the world fundamentally limits this work and moreover prevents even a discussion about the needs of such underrepresented regions and cultures as it relates to automatic 3D garment representation and generation.

**Strengths And Weaknesses:**

Strengths:

In my evaluation, the proposal of this paper is very good. Indeed, representation of 3D garments is a challenge and this paper provides an interesting sewing pattern based representation. The authors also do a great job of highlighting various design choices involved in both encoding and decoding this representation and thus building their proposed – Neural Sewing Machine. In particular, I believe the following are the set of strengths of this paper:

1. Unified Sewing Pattern Encoding: I like this encoding scheme as it allows for representing clothes of diverse categories, shapes and topologies. The fact that the authors chose PCA to come up with the encoding is again very interesting – something well suited for this problem setting as there is only so much variety in clothing.

2. 3D Garment Decoding: The authors provide a principled approach to decode the encoding by learning a 2D UV map and a mask map which together represent the various panels.

3. Structure Preserving losses: The proposed losses are intuitive and crucial for getting good results. The ablation that the authors provided validates their importance.

4. Mean Geodesic Length Error: I particularly liked this metric to evaluate the quality of topology and the authors show that their method gives good results on this metric.

Before I mention the weaknesses of the paper, I want to emphasize that the authors have presented an interesting method to model and generate 3D garments and I personally feel that the idea has a lot of potential. However, I think that authors have left out several important experiments and their analysis does not adequately back the claims they are making. Moreover, I don’t find their choice of baselines very convincing. I also find the paper is hard to read in several places with figures missing captions/ not being clear enough.

1. I think the authors should do a better job at describing the related work. They mention them all in a single line L112, L115 but that’s pretty much what the reader gets. At the present moment, it is hard to properly contextualize the contribution of this paper with respect to existing literature.

2. The authors show comparison with general purpose and ‘freeform’ baselines such as pixel2mesh and anchorUDF which is good to know but by no means is a fair comparison in my opinion. Such weak baselines tell little about the efficacy of the proposed method. The authors dedicate half a page in the related work on template-based models and garment structure modeling – which clearly shows that there are plenty of closely related recent papers (with open source code) that specifically target the garment generation problem. However, the authors don’t provide any comparisons for these. Therefore I am not convinced that this paper is really state of the art (not that it matters).

3. I am also curious to see an ablation for the UV position maps as an intermediary representation. Particularly, how well does it compare against those approaches that used an alternative image based representation.

4. I think the paper would benefit if authors spend some time in making the figures more explanatory. For example, in Fig 6 there are no captions as to which one is the prediction and which is ground truth. Moreover, I qualitatively cannot make sense what are the artifacts manifested by each loss term ablation. Perhaps an inset focusing on the artifact will help.

5. Novelty on surface normal loss – I think authors should not claim novelty on this one as it is pretty much well known in the area of 3D reconstruction.

6. While the authors introduce a new metric – Mean Geodesic Length Error (MGLE) to measure the quality of topology, I believe that the authors should have provided more analysis for the behavior of this metric such as the effect of the number of samples K. This is crucial because this metric seems to be important to show the superiority of the proposed method and I suspect that for different values of the sample K, the results will be very different. I hope to see results with a much larger value of K (on the order of a few 1000s atleast).

---

> ### Author Response · Authors · 2022-08-02
> ****Response to Reviewer HS7n** (part 3)**
>
>
>
> > Concern 9: In the dataset, the authors say L245 -- "We use the first 90\% part of the data for each base type … and remaining as the test set." Where is the validation set? If the authors fine-tuned their model on the test set, then this basically throws all of the experimental evaluation into jeopardy.
>
> **A9:** Thanks! We don't use the test set for searching hyperparameters or fine-tuning our model. We train and adopt our model based on the recovered garments in the training set and the convergence of the training loss. This is standard practice in common deep learning. For example, on ImageNet (and CIFAR and MNIST), only training and validation sets (sometimes called test sets) are currently used, and no distinction is made between test and validation sets.
>
> > Concern 10: I also didn't understand the section on controllable garment editing. It looks interesting, but I could not parse through the procedure involved in getting that.
>
> **A10:** Thanks for your interest! Controllable garment editing aims at editing a 3D garment. We first train PointNet[9] to predict a latent embedding from the 3D garment points, from which the sewing pattern is recovered by PCA inverse transform. Then we can edit the panel shapes/categories by changing the panel shapes or adding/deleting panel types. Finally, the edited sewing patterns are fed into our NSM garment decoder to obtain the edited 3D garments. The introduction of the sewing pattern makes our NSM capable of controllable 3D garment shape editing under geometry and topology variation. We added more explanations and demonstrations in the revision to make this part more readable.
>
> > Concern 11: I am surprised that the authors didn't talk about the obvious limitations of their work, i.e., both technical and in terms of its social impact.
>
> **A11:** Thanks! We will discuss more limitations (including your concerns) in the revision.
>
> > Concern 12: Due to the choice of their representation, I doubt that their method generalizes to other clothing styles. The authors neither experiment with the generalization nor provide any comments on it.
>
> **A12:** Many thanks for this creative question! Our NSN **can generalize** to other clothing styles due to two advantages. **First**, combining the basic panel groups may result in new garment categories not shown in the training set. **Second**, introducing the basic panel group also helps our NSM to generalize to novel garment categories. Specifically, a new garment category can be decomposed into basic panel groups, and if some basic panel groups are not included in the training set, we can append them into our training set and fine-tune the model to predict the novel category. For example, a 3D T-shirt with a pocket can be appended to the training set by modeling the basic panel group of the pocket panel. We will add this discussion in the revision.
>
> > Concern 13: The dataset used in this paper contains clothing styles that are not very inclusive, e.g., excluding South Asian clothes.
>
> **A13:** Thanks for this imaginative comment! It is still difficult for our method to model 3D garments with irregular sewing patterns, as the irregular panels have unknown vertex and edge numbers and are hard to be captured by the PCA algorithm. How to parameterize irregular garments via a unified representation is a challenging and important problem; we will explore this in future works.
>
> [1] Ma et al. "Learning to dress 3d people in generative clothing.", 2020.
>
> [2] Bhatnagar et al. "Multi-garment net: Learning to dress 3d people from images.", 2019.
>
> [3] Jiang et al. "Bcnet: Learning body and cloth shape from a single image.", 2020.
>
> [4] Hong et al. "Garment4D: Garment reconstruction from point cloud sequences.", 2021.
>
> [5] Korosteleva et al. "Generating Datasets of 3D Garments with Sewing Patterns.", 2021.
>
> [6] Yao et al. "Quasi-Newton solver for robust non-rigid registration.", 2020.
>
> [7] Wang et al. "Pixel2mesh: Generating 3d mesh models from single rgb images.", 2018.
>
> [8] Zhao et al. "Learning anchored unsigned distance functions with gradient direction alignment for single-view garment reconstruction.", 2021.
>
> [9] Qi et al. "Pointnet: Deep learning on point sets for 3d classification and segmentation.", 2017.

---

> ### Author Response · Authors · 2022-08-02
> ****Response to Reviewer HS7n** (part 2)**
>
>
>
> > Concern 5: Authors should not claim novelty on surface normal loss as it is pretty much well known in the area of 3D reconstruction.
>
> **A5:** Thanks for pointing that out! We adjusted our statement to avoid misunderstanding as we only introduced the surface normal loss for learning the 3D garment deformation.
>
> > Concern 6: The authors should have provided more analysis for the behavior of MGLE, such as the effect of the number of samples K. I suspect that for different values of sample K, the results will be very different. I hope to see results with a much larger value of K (on the order of a few 1000s at least).
>
> **A6:** Thanks for this insightful suggestion! As advised, we conduct sensitivity analysis to K in the MGLE metric. We use the task of 3D garment reconstruction from the sewing pattern for validation. The table below shows that the performance is not sensitive to K. Moreover, we can see that K greatly affects the evaluation time. For example, when K=1,000, the evaluation time is already more than 10 hours. Hence, we didn't provide the results of Ks greater than 1,000 due to the computational cost and unnecessity.
>
>
>
>
> | K  | Time(h) | MGLE |
> |  ----  | ----  |  ----  |
> | 2  | 1.38 | 3.52 |
> | 10  | 1.8 | 3.53 |
> | 20  | 2.3 | 3.54 |
> | 40  | 3.6 | 3.53 |
> | 80  | 7.6 | 3.52 |
> | 1000  | 10.4 | 3.51 |
>
>
> > Concern 7: L59: "Our NSM guarantees …" -- I don't understand how this statement is true. It's not that the 2D UV map is an isometric parameterization of any kind. Maybe I am missing something?
>
> **A7:** Thanks for this insightful question! Generally, a 2D UV map is not an isometric parameterization of 3D shapes in general scenes (e.g., maps or globes). But Garments have a prior that is different from general scenes, i.e., garments are constructed by cutting the panels from flat fabric and then sewing them into 3D space. Hence, 2D UV maps and sewing patterns are closely isometric parameterizations of the 3D garments when there are limited stretches on the 3D garment. Such priors are reasonable as we usually include these priors for face and human body reconstruction.
>
>
> > Concern 8: I have absolutely no clue how using physically based simulators for interacting with the clothes is a “contemporary line of work” to the proposed method, as the two are completely different items. What's the context or argument here? Also, the authors claim that their method creates garments that can be manipulated but nowhere do they qualify or provide any evidence for this one.
>
> **A8:** We really appreciate your profound insights! We make comments about physics-based simulators for two reasons. **First**, we refer to physics-based simulators to show that our NSM is more computationally efficient than physics-based simulators. Precisely, physical-based simulators require about 20~30s for a sample, while our NSM takes about 0.6s for each sample. **Second**, we highlight physical-based simulators because they can generate more robust and realistic results than geometry-based methods and can be used to correct the artifacts introduced by geometry-based methods. Our NSM can also benefit from physical-based simulators (see A3 to Reviewer 1Ro2).

---

> ### Author Response · Authors · 2022-08-02
> ****Response to Reviewer HS7n** (part 1)**
>
>
>
> We believe that Reviewer HS7n is an **excellent** reviewer who, while raising many concerns, also used a long text to appreciate the strengths of our article. Thank you so much for your valuable suggestions! Please see our response to your concerns below.
>
>
> > Concern 1: The authors should do a better job at describing the related work. They mention them all in a single line L112, L115, but that’s pretty much what the reader gets. At the present moment, it is hard to properly contextualize the contribution of this paper with respect to existing literature.
>
> **A1:** Thank you for this valuable suggestion! We clarify L112 and L115 here and add more discussion \& comparison to the related work in the revision. Specifically, [1,2] model 3D garments as the displacement on the SMPL model, but they have limitations in modeling loose garments with large wrinkles, for garment folding may cause one-to-many mapping between SMPL models and garment surfaces. Another line of works [3,4] registers garments with different shapes and topologies to a template mesh with a fixed vertex number and topology. These garments with a unified representation are then mapped to low-dimensional representations that are used for reconstructing 3D garments from images/videos. But these methods have limitations in representing garments with diverse variations because large garment shape variation is non-linear even on a T-pose human body and thus could not be expressed by a template with a fixed number of vertices.
> Also, the large wrinkles on the garment surface make the registration not robust for loose garments, which may cause the incorrect garment topology after registration. In contrast to the above works, sewing patterns describe the intrinsic structure of garments, which is capable of representing garments with diverse shapes.
>
>
> > Concern 2: The authors show a comparison with the general purpose and "freeform" baselines such as pixel2mesh and anchorUDF, which is good to know but by no means is a fair comparison, in my opinion. Such weak baselines tell little about the efficacy of the proposed method.
>
>
>
> **A2:** Thanks for this critical advice! As suggested, in this rebuttal, we compare our NSM with BCENet, a state-of-the-art template-based model on the garment3D dataset [5]. This implementation took up a lot of our rebuttal time. Specifically, we register the garments in each category to a template garment mesh using the open-source non-rigid registration method [6]. After registration, we use PCA to map the garment meshes to a subspace and obtain a low-dimensional representation for each garment. Then we train one network to predict the garment category and another to predict the low-dimension representation from an image, from which a 3D garment is recovered with the inverse PCA transform. We evaluate the accuracy of the predicted garment geometry and topology with the evaluation metrics of Chamfer distance, P2S, and MGLE. The results in the table below show that our NSM far outperforms BCENet on the two metrics Chamfer and P2S and is on par with BCENet on MGLE. These comparisons demonstrate that our NSM outperforms template-based BCENet on reconstruction tasks with diverse shapes and that some template-based methods may also have a particular structure-preserving ability but at the cost of losing muti-shape reconstruction robustness.
>
>
> | Methods  | Chamfer | P2S | MGLE |
> |  ----  | ----  |  ----  |  ----  |
> | Pixel2Mesh[7]  | 5.23 | 3.84 | 10.81 |
> | AnchorUDF[8]  | 3.31 | 4.17 | 4.61 |
> | BCNet[3]  | 4.69 | 4.33 | 3.42 |
> | NSM  | 2.08 | 1.90 | 3.73 |
>
>
> > Concern 3: I am curious to see an ablation for the UV position maps as an intermediary representation. How well does it compare against those approaches that used an alternative image-based representation?
>
> **A3:** Many thanks! This ablation has been in Section 4.1.
> Specifically, our NSM is the method that **uses the UV position maps as an intermediary representation** because it first predicts the latent embedding and panel classes from images and then decodes the latent embedding to the UV position maps, from which the 3D garment is recovered. AnchorUDF is the method that **uses an alternative image-based representation** because it predicts the 3D garment shape as the implicit function from the corresponding image-based representation without modeling the UV position maps. As shown, using the UV position maps as an intermediary representation significantly outperforms using an alternative image-based representation.
>
>
> > Concern 4: The paper would benefit if the authors spent some time making the figures more explanatory (e.g., Figure 6).
>
> **A4:** Thank you for your careful reading and advice! We made Fig 6 more explanatory and readable by adding more indicators and zoom-in to identify each sub-image and the artifacts in the revision.

---

> ### Comment · Reviewer_HS7n · 2022-08-08
> **Response to author rebuttal**
>
> The authors have tried to address most of my concerns and I appreciate the hard work that they have put forth in coming up with the rebuttal. I am satisfied with their response and have upgraded my score. I still feel that the paper in its current form falls short of being a solid contribution to the community owing to the limitations I and other reviewers have raised. Nevertheless, the authors propose a clear hypothesis and solution and provide sufficient validations to back up their claims.

---

### Official Review · Reviewer_1Ro2 · 2022-07-10

**Rating:** 6
**Confidence:** 4
**Soundness:** 3 good
**Presentation:** 3 good
**Contribution:** 3 good

**Summary:**

This work focuses on modeling 3D garments with a novel neural sewing machine (NSM).  The NSM is a learning framework that can effectively encode garment representations, which could facilitate 3D garment construction and manipulation.  Firstly, it utilizes a unified sewing pattern encoding module to embed sewing patterns.  Then, the embeddings are decoded into a 3D garment by using the UV-position maps with masks.  A few loss terms are introduced to preserve the inner-panel structures of the 3D garments.  Extensive experiments validate its superiorities over previous SoTA methods.

**Questions:**

Generally, I think it is a nice paper, and maybe the authors can provide more real-captured images as I mentioned in the ``Weakness''.

**Limitations:**

The manuscript has a section to discuss the limitations and broader impact.  But the method limitations have not been properly discussed.

**Strengths And Weaknesses:**

Strengths:
1. Using sewing patterns for 3D garment modeling is a good idea and also a nice contribution, which supports both 3D garment reconstruction and controllable editing.
2. Extensive experiments show that the proposed method outperforms previous SoTA methods quantitatively and qualitatively.

Weakness:
1. This manuscript claims that previous methods failed to learn reconstructable and manipulable representations with complicated topology.  Maybe a few more examples with complicated or straight shapes can be provided.

2. Also, all the experiments are conducted on synthetic data, and it would be nice if real-captured or even in-the-wild images can be used for evaluation.

3. It seems like this work can only reconstruct the garments with the canonical pose, and many local details (e.g. wrinkle) are missing even if they can be observed in the input image.  And the possible penetrations between the reconstructed 3D garments and the 3D human body are not considered during reconstruction and manipulation.

---

> ### Author Response · Authors · 2022-08-02
> ****Response to Review 1Ro2****
>
>
>
> We thank reviewer 1Ro2 for the thoughtful comments. We are encouraged that the reviewer found our approach valuable to the community. Please see our responses below.
> > Concern 1: Maybe a few more examples with complicated or straight shapes can be provided.
>
> **A1:** Many thanks for this valuable suggestion! As suggested, **we added more examples with complicated and straight shapes to Figure 7 in the revision**. For example, the jacket in the second row has a more complex structure than the jumpsuit in the third row. These examples demonstrate that our NSM can better preserve the garment topologies and maintain detailed 3D garment shapes, especially for loose garments with large wrinkles.
>
> > Concern 2: It would be nice if real-captured or even in-the-wild images could be used for evaluation.
>
> **A2:** Thanks a lot! As advised, **we used the real-captured in-the-wild images for evaluation**. Given an image of a T-pose person, we first estimate the camera parameters using [1] and a 2D cloth semantic segmentation using [2], then we fit the trained NSM to the image to obtain the 3D garment. We set the input embedding for the NSM decoder as learnable variables and fixed the NSM decoder parameters. We optimized the projection of the predicted garment to match the cloth segmentation on the image. The results will come soon these days because we spent lots of time running the experiments posed by Reviewer HS7n in the past week. (We have updated the visualization results in Sec5 of our revised supplementary)
>
>
> > Concern 3: This work can only reconstruct the garments with the canonical pose ... and many local details (e.g., wrinkle) are missing ... the possible penetrations between the reconstructed 3D garments and the 3D human body are not considered ...
>
> **A3:** Many thanks!
> **First**, the reason why we only reconstruct garments with the canonical pose is attributed to benchmark availability, i.e., existing sewing-pattern benchmarks only contain canonical-pose data. We will expand NSM to multi-pose bodies in our future work, dependent on the benchmark availability. One potential solution is to use "virtual bones" techniques (see [3]) for modeling loose garment deformation, which is beyond the scope of this paper. **Second**, the reason why wrinkles are missing is attributed to the complexity of loose garment deformation, which is more than driven by the human pose or shape. One solution is to model the detailed wrinkles with a generative model (see [4]). **Third**, to address possible penetration between a garment and a human body (and avoid missing details), we introduce a post-process by simulating the garment with the human body via MAYA in this rebuttal. Note that this differs from a pure physical-based simulation(PBS) in two folds. On the one hand, thanks to the good representations by our NSM, our post-process is about 30 times faster than purely PBS (i.e., 1s vs. 30s, see [5]). On the other hand, our NSM explicitly models the consistency between sewing patterns and the 3D garment surfaces to preserve garment topologies, allowing rich downstream tasks in computer vision and computer graphics. The qualitative results are shown in Figure 3 in our revised supplymentary.
>
> > Concern 4: The method limitations have not been properly discussed.
>
> **A4:** Thanks! We will discuss more limitations (especially your concerns) in the revision.
>
> [1] Kocabas et al. "Vibe: Video inference for human body pose and shape estimation.", 2020.
>
> [2] Gong et al. "Graphonomy: Universal human parsing via graph transfer learning.", 2019.
>
> [3] Pan et al. "Predicting Loose-Fitting Garment Deformations Using Bone-Driven Motion Networks.", 2022
>
> [4] Zhu et al. "Detailed wrinkle generation of virtual garments from a single image.", 2021
>
> [5] Korosteleva et al. "Generating Datasets of 3D Garments with Sewing Patterns.", 2021

---

### Meta-Review · Area_Chair_1teK · 2022-08-20

**Recommendation:** Accept
**Confidence:** Certain

**Metareview:**

This paper was reviewed by four experts in the field. Based on the reviewers' feedback, the decision is to recommend the paper for acceptance to NeurIPS 2022.

The reviewers did raise some valuable concerns that should be addressed in the final camera-ready version of the paper. For example, 1) the evaluation on real-world datasets can be incorporated, 2) more discussion can be added on the reconstruction of garments with non-canonical poses. The authors are encouraged to make the necessary changes to the best of their ability. We congratulate the authors on the acceptance of their paper!

**Award:**

No

---

### Decision · Program_Chairs · 2022-09-14

Accept